# Adaptation of the Health Literacy Survey Questionnaire (HLS_19_-Q) for Russian-Speaking Populations—International Collaboration across Germany, Israel, Kazakhstan, Russia, and the USA

**DOI:** 10.3390/ijerph19063572

**Published:** 2022-03-17

**Authors:** Maria Lopatina, Eva-Maria Berens, Julia Klinger, Diane Levin-Zamir, Uliana Kostareva, Altyn Aringazina, Oxana Drapkina, Jürgen M. Pelikan

**Affiliations:** 1Department of Public Health, National Medical Research Center for Therapy and Preventive Medicine, Ministry of Health, 101000 Moscow, Russia; ms.lopatina@gmail.com (M.L.); drapkina@bk.ru (O.D.); 2School of Public Health, Bielefeld University, 33501 Bielefeld, Germany; 3Institute of Sociology and Social Psychology, University of Cologne, 50923 Köln, Germany; klinger@wiso.uni-koeln.de; 4Department of Health Education and Promotion, Clalit Health Services, University of Haifa School of Public Health, Tel Aviv 6209804, Israel; diamos@zahav.net.il; 5Nancy Atmospera-Walch School of Nursing, University of Hawaiʻi at Mānoa, Honolulu, HI 96822, USA; uliana@hawaii.edu; 6Caspian International School of Medicine, Caspian University, Almaty 050000, Kazakhstan; altyn.aringazina@gmail.com; 7Austrian National Public Health Institute, 1010 Vienna, Austria; juergen.pelikan@goeg.at

**Keywords:** health literacy, HLS_19_-Q, questionnaire adaptation, Russian-speaking population, immigrants/migrants, cultural appropriateness, cultural responsiveness

## Abstract

The Russian language is the eighth most spoken language in the world. Russian speakers reside in Russia, across the former Soviet Union republics, and comprise one of the largest populations of international migrants. However, little is known about their health literacy (HL) and there is limited research on HL instruments in the Russian language. The purpose of this study was to adapt the Health Literacy Questionnaire (HLS_19_-Q) developed within the Health Literacy Survey 2019–2021 (HLS_19_) to the Russian language to study HL in Russian-speaking populations in Germany, Israel, Kazakhstan, Russia, and the USA. The HLS_19_-Q was translated either from English or from a national language to Russian in four countries first and then critically reviewed by three Russian-speaking experts for consensus. The HLS_19_ protocol and “team approach” method were used for linguistic and cultural adaptation. The most challenging was the adaptation of HLS_19_-Q questions to each country’s healthcare system while general HL questions were flexible and adaptable to specific contexts across all countries. This study provides recommendations for the linguistic and cultural adaptation of HLS_19_-Q into different languages and can serve as an example of international collaboration towards this end.

## 1. Introduction

Health literacy (HL) is globally recognized as a critical determinant of health [1]. According to the World Health Organization (WHO) Health Promotion Glossary, health literacy represents the personal knowledge and competencies that accumulate through daily activities, social interactions, and across generations [2]. Personal knowledge and competencies are mediated by organizational structures and the availability of resources that enable people to access, understand, appraise, and use information and services in ways that promote and maintain good health and well-being for themselves and those around them [3]. The HL definition, which was used in HLS_19_, determines HL as people’s knowledge, motivation, and competencies to access, understand, appraise and apply information to make judgments and make decisions in everyday life concerning healthcare, disease prevention, and health promotion [4]. HL depends on the individual skills and abilities of information-seeking, decision-making, problem-solving, critical thinking, and communication [5].

Cultural factors play an important role in health literacy. WHO recognizes the importance of cultural factors and their impact on HL and health behavior [6]. Cultural factors can influence how health information is processed and how health-related decisions are made [7]. Since HL is the interplay of individual skills, circumstances, situational and social demands that change over time, people may be considered health-literate in one country but not in another [8].

Migration experience and immigration status as well as national language proficiency play an important role in HL across multiple countries [4,7,9]. Linguistic and cultural barriers can cause a low level of integration within the society and lead to limitations in terms of developing individual HL in the host country [10]. In turn, language may lead to stigmatization of migrants (immigrants) and to health inequity in the host society [11,12,13]. Cultural factors and experiences from migrants’ native countries may also affect HL in their host countries [9,10,11].

The Russian language is the eighth most spoken language in the world [14]. Russian speakers reside primarily in Russia and across the former Soviet Union (FSU) republics. For example, in Kazakhstan, the Russian language is the second official language [15]. Russian speakers comprise one of the largest populations of international migrants estimated at 25–30 million [7]. Alone, Russia (10.5 million) is the fourth source of international migrants in the world after India (17.5 million), Mexico (11.8 million), and China (10.7 million) [16].

Following the Soviet Union’s dissolution in 1991, many people relocated between the former republics, as well as moved internationally, primarily to the USA, Germany, and Israel [7,17]. FSU migrants to Germany also included many Resettlers (in German: (Spät) Aussiedler) or ethnic Germans whose ancestors started emigrating to the Russian Empire as early as the 1700s [18]. FSU immigrants have a common language (Russian), historical background, and come from similar Soviet or post-Soviet healthcare systems, notably different from the systems of their host countries [19].

Germany has an estimated 3.5 million FSU immigrants [20]. In Israel, there are almost 1.3 million FSU immigrants, representing more than a third of all Israeli immigrants [20]. In the USA, approximately 3–5 million people claim ancestry from the FSU region, which includes both recent immigrants and descendants of immigrants, and almost 1 million speak Russian at home [21].

The first assessment of HL among Russian speakers was completed in Israel in 2012 for their Russian-speaking migrant population as part of the European Health Literacy Survey (HLS-EU) [17]. The second was completed in Kazakhstan in Kazakh and Russian languages within the Asian HL survey in 2013–2014 [19]. Since then, the questionnaire has been updated from HLS-EU-Q to HLS_19_-Q. While the HLS-EU-Q was translated into Russian, there is no evidence regarding whether it was linguistically and culturally adapted for the Russian-speaking population.

In 2018, the WHO Action Network on Measuring Population and Organizational Health Literacy (M-POHL) was established, aiming to carry out periodic, high-quality HL surveys and support the collection of data on HL, as prerequisites for evidence-based policy and practice [22]. Within the M-POHL Network, the international research consortium developed a new questionnaire in English (HLS_19_-Q) based on the previous HLS-EU-Q. The instrument was then translated by countries participating in the Health Literacy Survey of 2019–2021 (HLS_19_). For 16 out of the 17 participating countries (all except Ireland), translation from the original English questionnaire into the local language(s) was required [23].

In this study, we highlight the importance of linguistic and cultural adaptation of the HL assessment instrument, which is usually omitted in population studies. The purpose of this study was to adapt the HLS_19_-Q for the Russian-speaking population in Germany, Israel, Kazakhstan, Russia, and the USA, based on the international collaboration of the M-POHL Network, and to provide a blueprint for the process of adapting the HLS_19_ questionnaire to other languages.

## 2. Materials and Methods

### 2.1. HLS_19_ Questionnaire

In the HLS_19_ questionnaire, general HL is measured with the instrument that was validated in four versions: 12, 16, 22, or 47 general HL items (HLS_19_-Q12, HLS_19_-Q16, HLS_19_-Q22 HLS_19_-Q47). These four versions were developed based on the HLS-EU-Q47 instrument, the conceptual framework, and the definition of HL used in the HLS-EU model. The short form with 12 items (HLS_19_-Q12) was developed and validated specifically for HLS_19_ [24]. Countries that preferred to simultaneously measure via the items in both the HLS_19_-Q12 and the HLS_19_-Q16, used a set of 22 items, enabling them to construct both instruments, mainly for comparison with previous studies. The long HLS_19_-Q47 version of the instrument for measuring general HL was adapted in Germany and Kazakhstan. The HLS_19_-Q22 version was adapted in Israel and Russia and the HLS_19_-Q12 version in the USA. All of the mandatory 31 correlates on determinants and consequences of HL were adapted in Israel, Kazakhstan, and Russia, most in Germany (25 out of 31), and several in the USA (7 out of 31) (see Appendix A). These included demographic determinants (gender, age), socio-economic determinants (education, subjective social status, financial deprivation, employment status, migration status), and other determinants (training in a health care profession, social support). Consequences of HL were comprised of health behavior and lifestyles (body mass index, physical activity, smoking behavior, alcohol consumption, fruit and vegetable consumption); health status included self-reported health, chronic illness, and limitations of daily activities caused by health problems. Health care utilization was comprised of emergency service calls, general practitioner/family doctor, medical or surgical specialist visits, inpatient and day-patient hospital service use.

### 2.2. Methods of HLS_19_-Q Adaptation

According to the HLS_19_ study protocol, the requirement regarding the questionnaire translation procedure into the native language(s) of each participating country included four steps [23]:(1)Two forward translations;(2)Comparison of two translations and decision on the most appropriate translation in the case of differences, based on expert consensus;(3)Final check of comprehensibility of the translated version through a focus group discussion with participants similar to potential survey respondents;(4)Pretest of the translated national HLS_19_-Q version based on a field-test including at least 30 interviews.

HLS_19_ study protocol also recommended using purposeful sampling for the field-testing to ensure equal distribution of participants in terms of age, gender, and education. There were no specific standards by the HLS_19_ International Coordination Center (ICC), for the questionnaire adaptation for countries with different languages spoken, other than their official language. Since there were several participating countries where the Russian language is spoken (Germany, Israel, Kazakhstan, Russia, and the USA), which is very culture- and context-specific, researchers from these countries initiated and consolidated their efforts in a “team approach” [25] for adaptation of the questionnaire into the Russian language.

The process of the HLS_19_-Q adaptation included a review of HLS-EU-Q as a predecessor of HLS_19_-Q and HLS_19_-Q general HL items and HL correlates. The steps of the HLS_19_ questionnaire adaptation are presented in Figure 1. At the first step (1), item re-wording and adding or removing items and examples was done. At the second step (2), translation and adaptation of the English language main HLS_19_- into national languages in Russia and Germany was carried out. In Russia, the process of adaptation of the HLS_19_ questionnaire into Russian was performed by the National Study Center (NSC) and included two forward translations, a back translation, comparison of translations and decisions of NSC, focus group, expert review, and field-testing. In Germany, two forward translations into German were carried out: one by the data collection agency (DCA), one by NSC. Then a comparison of the translations was undertaken and consensus with DCA was reached. This version was pretested in the required field-test. Additionally, consensus with the Austrian and Swiss NSC, who used German-language questionnaires as well, was achieved. At the third step (3), the process was finalized, which included discussions with participating NSCs based on experiences and recommendations from German and Russian teams. At the fourth step (4), changes were made in the pre-existing translations in Russia and Germany. In Israel, the adaptation process into Hebrew and Arabic included one forward translation and one back translation, managed by the NSC to Hebrew. In Kazakhstan, the questionnaire was translated by NSC from English into Russian (and Kazakh) and back-translated. It was then reviewed and discussed with the Russian study team until consensus was reached. Finally, at the fifth step (5), translation and adaptation for Russian-speaking (im)migrants were carried out. These steps are described in more detail in the following sections.

### 2.3. Adaptation of the HLS_19_ Questionnaire for Russian Speakers

In all participating countries, the adaptation process was also associated with the mode and time of data collection (see Appendix A).

#### 2.3.1. Adaptation in Russia and Kazakhstan for Russian-Speaking Population

In Russia, professional translators performed two direct and back translations in January 2019, after which the research group compared the variants with minor differences identified and discussed until full concordance was achieved. In August 2019, a focus group was carried out among 12 laypeople without formal medical education, selected by purposeful sampling to ensure even and heterogeneous distribution by age, gender, education, and place of residence (rural and urban). The questions were read by the interviewer, and for the convenience of respondents, an individual paper form was developed where respondents highlighted unclear questions and recorded their comments. The interview was then followed by a discussion among participants. When the focus group was conducted, the Russian-speaking expert from the USA was invited to visit Russia to participate and contribute to the organization and data analysis of the focus group study. At the next step, the expert opinion method was applied via consensus among eight public health experts. All experts unanimously agreed to use the shorter version of the questionnaire (HLS_19_-Q22) and add 7 questions from the full version (HLS_19_-Q47) and totaling 29 core HL questions. Subsequently, the field-testing of the questionnaire with 29 core HL items was carried out to check the usability and feasibility of the questionnaire. The feedback from interviewers (N = 10) and respondents (N = 80) clarified answer categories, checked for question understanding, the overall design of the tool, understandability of the written instructions for interviewers, and evaluated the duration of a survey per respondent. There were no difficulties in the use or understanding of questions identified for either interviewers or respondents. The HL study in Russia was conducted among adult residents of 18 years and older in three regions (Novosibirsk, Karelia, and Tatarstan) using the method of personal paper assisted interviews (PAPI) in households from November to December 2019 using the HLS_19_-Q22 and 7 additional items (see Appendix A).

In Kazakhstan, the questionnaire was translated by NSC from English into Russian and back-translated. It was then reviewed and discussed with the Russian study team until consensus was reached. Due to the COVID-19 pandemic barriers, the survey could not be carried out as planned in Kazakhstan.

#### 2.3.2. Adaptation in Germany and the USA for Russian-Speaking Migrants

First, the German version of the original English HLS_19_ questionnaire was translated into Russian by two professional translators, who were contacted by the DCA in March 2020. Then, three native Russian speakers (one from Russia, one from Kazakhstan, one from the USA) with HL expertise from different countries performed four rounds of reviews. The two variants of translations by professional translators were critically reviewed by HL experts, compared to their local translations, and revised accordingly by the German team. Each expert provided insights in English to the German research team as this was the common language. The questionnaire was evaluated and edited until full concordance was reached and approved. Figure 2 shows the adaptation process in Germany. In June 2020, the Russian language questionnaire was pretested in a pilot study under real conditions with one male and four female immigrants from Kazakhstan, Russia, Latvia, and Ukraine who were between 45 and 63 years old and had been living in Germany for 23–28 years. The HL study was conducted among adult Russian-speaking FSU immigrants, including those who migrated or whose parents migrated, and used the PAPI method with bilingual interviewers in August–September 2020 (see Appendix A).

In the USA, the same HLS_19_-Q in Russian was used as the one in Germany. One of the HL experts who provided a review of the translated German survey conducted the USA study. The USA HL study assessed HL among Russian-speaking (im)migrants who had emigrated at ages of 14 years and older via an online survey (available to respondents from any state) in September–October 2020. The short 12 question version of HLS_19_-Q and only a selected number of correlates were used (gender, country of origin, education, social status, employment, problems with paying bills, health status) (see Appendix A). The shorter version of the survey was used in order to adapt to the online format and in accordance with available resources, as it was not a nationally-funded study.

#### 2.3.3. Adaptation in Israel for Russian-Speaking Migrants

In January 2019, the English version of the HLS_19_-Q22 questionnaire was translated professionally into Russian, Arabic, and Hebrew. Regarding the translated Russian version, the questionnaire was back-translated to Hebrew by a health professional and compared to the validated Hebrew version, which had already undergone back translation and for which a pretest had already been conducted (see Figure 1 step 5). In addition, several questions were added to all three languages versions that were not included in the international version. The HL study was carried out among the Russian-speaking population using computer-assisted web interviews in December 2020–January 2021 (see Appendix A).

## 3. Results and Discussion

In Germany, the German version of HLS_19_ was used for the Russian translation as some cultural adjustments to the original English version were already made to better suit the German context (e.g., instead of “neighborhood”, which in the direct German translation can also mean the people living in the neighborhood, “residential area” (Wohnumgebung) was used). The feedback from the pretest in Germany mostly concerned linguistic aspects. A challenge was encountered in translating terms for German health providers, e.g., general practitioner (Allgemeinmediziner) or family doctor (Hausarzt). The pilot study participants were longtime residents of Germany and were more familiar with the German terms; thus, it was decided to include the German terms in brackets following the Russian terms as recommended by HL experts.

In Israel, several discrepancies were evidenced, for example, “health literacy” was back-translated to “medical literacy”, and “commercial interests” was back-translated to “marketing interests”. These issues were clarified among those involved in the translation processes including the principal investigator of the Israel survey.

In Russia, as a result of the focus group, changes were made in accordance with the clear language principle. Out of 47 questions, 6 were difficult to understand at the first reading, 10% were incomprehensible to the respondents, 5% seemed redundant and repetitive, and in general, all respondents noted that the questionnaire was too long. The proposed changes were discussed in the international M-POHL research consortium meeting, accepted by experts, and also incorporated into the questionnaires which were later developed for the Russian-speaking population in participating countries. Examples of the input of the focus group comments during the development of the HLS_19_ questionnaire are introduced in Appendix B (cultural adaptation) and Appendix C (linguistic adaptation). An example of a cultural adaptation: the question “Do you have a family member or a friend to take with you to a doctor’s appointment?” caused confusion among Russian respondents because they could not understand the purpose of taking a family member or a friend to a doctor, since it is not a tradition in Russian culture. All Russian respondents understood it as a question of trust and suggested changing it to: “Do you have a family member or a friend to share information regarding personal health?” As a result of the experts’ discussion, it was moved to optional questions (Appendix B). A linguistic adaptation example: in question “How easy or difficult is it for you to judge which everyday behaviour is related to your health? (Instructions: drinking and eating habits, exercise, etc.)”, the word “behavior” was not clear and not usually used in relation to health. Suggestion for adaptation was to change “behavior” to “habits” and substitute “related to” for “affect” (Appendix C).

Overall, the undertaken adaptation can be conditionally classified into linguistic and cultural. In Russia, out of 90 questions, 86 (95%) were linguistically or culturally adapted. It is also important to highlight that we found the general questions of HLS_19_-Q to be flexible and adaptable to specific contexts of each country, which can be considered a strength of the questionnaire. The most challenging was the adaptation of HLS_19_-Q questions to each country’s healthcare system. This is important because depending on the method of data collection, there was or was not the possibility to include individual clarification in real-time (face-to-face interview vs. online survey).

Quite often, health questionnaires are developed and validated in one country or in one language and then used in different countries across various settings without careful consideration of contextual and linguistic differences [26]. Often, little attention is paid to cross-cultural adaptation [26]. The translation of a questionnaire without linguistic and cultural adaptation may lead to misinterpretations and unreliable data [27]. This study highlights the importance of instruments linguistically and culturally adapted for the local population, which is also in line with one of the major health literacy participatory approaches to involve the target group and experts in the process [4]. Although country-specific adaptation is desirable, it may not always be possible because of a lack of resources, time, and expertise [26].

The HLS_19_ protocol and the method of a “team approach” were employed to adapt the HLS_19_ questionnaire to each participating country [25]. The M-POHL recommends conducting an HL assessment of the population on a regular basis and the adapted versions of the questionnaire could be used in future assessments of the Russian-speaking population across participating countries. This process helped to find the balance between several translation variations and identified discrepancies between formal and informal language, literal versus understandable translations, and regional language and culture differences.

The fact that the process of the development of the HLS_19_ instruments was carried out within this international collaboration led to less complication in the local adaptation process. In the questionnaire design, especially in international (comparative) survey projects, it is important and necessary to collaborate with international colleagues and plan ahead when developing measurement instruments to avoid translation difficulties [28]. For the HLS_19_ study, researchers from participating countries were able to bring in their perspectives to ensure that the instruments were suitable for their respective national and cultural contexts. Because some countries began their translation of the original questionnaire and data collection earlier according to the timeline (e.g., Germany and Russia), minor changes were suggested to M-POHL, which allowed for the improvement of the questionnaire in general, and helped other countries that began their translation later. Furthermore, for reproducibility, it is critical to document the adaptation process and decisions for supporting national follow-up surveys as well as international adaptations and to help improve future international collaboration processes [29].

Health literacy is a complex concept, referring both to content and context [30]. Its measurement methods go beyond patients’ competencies in dealing with medical services, reading, and numeracy skills (functional, performance-based methods) and also capture interactive and critical health literacy in everyday life contexts [31]. Studies show that the way health literacy is understood and approached affects the way it is measured [30,31]. HLS_19_ instrument measures critical HL and is, therefore, more embedded into the context. This means the questions are more context-sensitive and one must be more careful with translation/adaptation than with functional HL measurements, i.e., the measurement approach affects the adaptation efforts.

Requests for the adapted instruments should be directed to the International Coordination Center of the M-POHL Network.

## 4. Conclusions

A substantial adaptation of the HLS_19_ questionnaire led to the development of four linguistically and culturally appropriate tools, relevant to the context of each country. Based on the obtained results, practical recommendations were outlined for researchers planning health literacy surveys through the HLS_19_ questionnaire. Based on the obtained results during this collaborative international multicultural study, the authors summarized a blueprint for researchers from other countries for adaptation of HLS_19_-Q in different languages:In addition to two forward and one back-translations, use a “team approach” as the best practice to contextualize survey questions to the national context. The case for conducting more than one forward translation is that if there are flaws detected, the back-translation will not serve its main purpose. Therefore, two forward translations are recommended;Choose professional translators who are familiar with the healthcare system, or have worked in health care, with experience in translating health-related materials;Conduct pretests and/or focus groups to evaluate survey questions;Invite experts in the field to offer their feedback about linguistic, stylistic, data collection methodology, and cultural adaptation categories;Engage international researchers to gain “outside the box” perspectives and gather insightful adaptation recommendations.

This questionnaire adaptation study can serve as an example of collaborative work and as an important step prior to and in preparation for a multi-cultural and multi-lingual survey.

## Figures and Tables

**Figure 1 ijerph-19-03572-f001:**
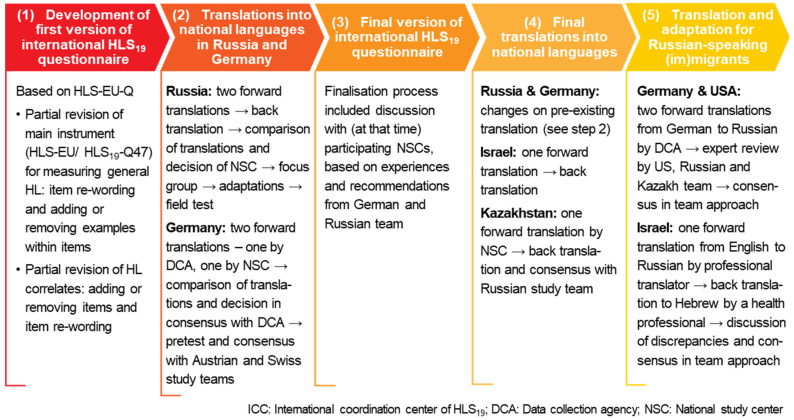
Steps of the HLS_19_ questionnaire adaptation.

**Figure 2 ijerph-19-03572-f002:**
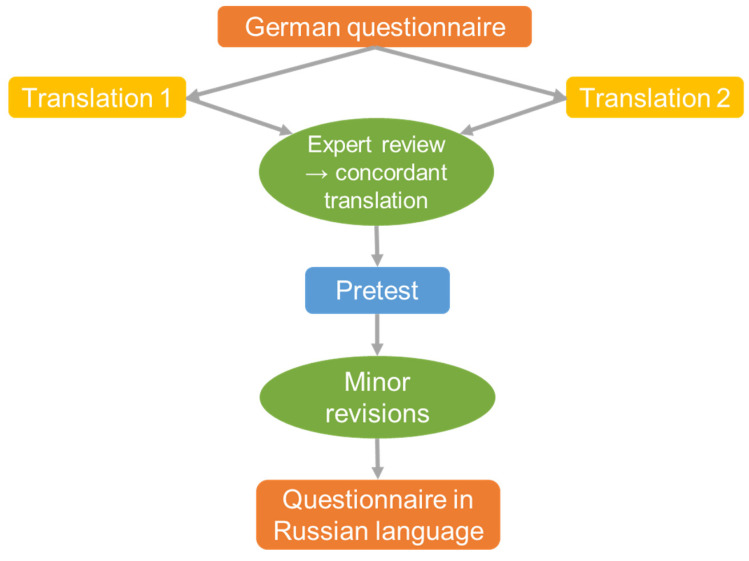
HLS_19_-Q adaptation process in Germany.

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
