# Peer review of "Adaptation of the Health Literacy Survey Questionnaire (HLS19-Q) for Russian-Speaking Populations—International Collaboration across Germany, Israel, Kazakhstan, Russia, and the USA"

_ijerph, 2022, doi:10.3390/ijerph19063572_

Round 1

Reviewer 1 Report

The article's topic is interesting and relevant: the Authors described the adaptation process of the HLS19-questionnaire for Russian-speaking populations.

I have a few recommendations in order to improve the manuscript:

The abstract is too general, the reader cannot get a clear picture of the adaptation process and the recommendations from the abstract. So it can be expanded with this information.

In the introduction, some paragraphs seem to be not connected to the main purpose of the article, like the relationship between HL and social determinants of health and cultural factors. It would be good to omit these paragraphs or rephrase them and show how these are connected with the adaptation process. In line 92, it is stated that little is known about the adaptation to new health care systems in host countries, but again, the importance of this from the point of view of the present study is questionable.

In the Methods, the Authors listed the different topics covered in the questionnaire, but some of them will be not discussed later, or it is not part of the HL questionnaire. Maybe it would be worth focusing only on those variables which were used during the adaptation process. There is a discrepancy between the text and the content of Figure 1 (e.g. the translation process in Germany), so this part should be revised.

There is some information (e.g. the translation process) described both in the methods and the results section, which is a little bit disturbing. The results section also contains some methodological information, like the description of the American questionnaire or the data in Table 1. The authors did not clearly distinguish what belongs to the used methods and the results. It would be worth thinking this over and rephrasing the text accordingly in order to have a clearer layout of the article. The content and purpose of Table 1 are not clear, on the one hand, because it contains methodological information, but it was placed in the results, on the other hand, because it is not discussed in the text. For instance, if the target population in Germany also contains people who speak only German, how can they help during the adaptation of the Russian version of the questionnaire?

From the beginning of the Discussion, the summary of the main results are missing, and the meaning of the last sentence (our next step is ….) is unclear. The purpose of the article was to describe the development of the Russian version of the questionnaire, then how can be the next step the same?

Minor corrections:

Line 126: maybe use demographic instead of socio-demographic for age and gender

In the results section, the monogram of the person responsible for a task is given, but it is not clear that this is the meaning of the abbreviation, for instance, in the case of the UK. Furthermore, the type of the used parenthesis is incongruent (both () and [] is used).

Reviewer 2 Report

The manuscript describes the process used to adapt the Health Literacy Survey Questionnaire (HLS19Q) for Russian-speaking populations. The paper is clear and well-written. The methodology is correct. A few comments to improve the manuscript.

Introduction

Line 57-61. A few references are outdated. For example, a recently published meta-analysis found that high health literacy is associated with higher uptake of cancer screening programs (DOI: 10.1016/j.ypmed.2021.106927). Also, health literacy seems to play a role in relation to vaccine hesitancy towards COVID-19 vaccination and adherence to preventive measures (doi: 10.17061/phrp30342012, doi: 10.1080/21645515.2021.1984123 and doi:10.1007/s12144-021-02105-8), confirming its importance in determining health behavior. These should be added.

Methods & results

Methods are clear. The use of figures and tables helps to understand the process used.

The authors should provide the adapted instruments as supplementary material.

Discussion

As pointed out by the authors in the introduction part, the health literacy concept is very complex. Its measurement methods can be broadly grouped into four categories according to the structure of the tool used: using word recognition items, using reading or numeracy comprehension items (performance-based), using self-reported comprehension items, or using a mixed method (i.e., combination of self-reported and reading or numeracy comprehension items). Several studies have shown that how you define health literacy affects how you measure it and therefore also your findings. This concept should be mentioned in your discussion in relation to the adaptation of the HLS19Q instrument.
